# Thermodynamic Cycle Concepts for High-Efficiency Power Plans. Part A: Public Power Plants 60+

**Krzysztof Kosowski** [1], **Karol Tucki** [2,*], **Marian Piwowarski** [1], **Robert Stępień** [1], **Olga Orynycz** [3,*], **Wojciech Włodarski** [1] and **Anna Bączyk** [4]

1   Faculty of Mechanical Engineering, Gdansk University of Technology, Gabriela Narutowicza Street 11/12, 80-233 Gdansk, Poland; kosowski@pg.gda.pl (K.K.); marian.piwowarski@pg.edu.pl (M.P.); rstepien@pg.edu.pl (R.S.); wwlodar@pg.edu.pl (W.W.)
2   Department of Organization and Production Engineering, Warsaw University of Life Sciences, Nowoursynowska Street 164, 02-787 Warsaw, Poland
3   Department of Production Management, Bialystok University of Technology, Wiejska Street 45A, 15-351  Bialystok, Poland
4   Department of Hydraulic Engineering, Warsaw University of Life Sciences, Nowoursynowska Street 159, 02-776 Warsaw, Poland; a.baczyk@levis.sggw.pl
*   Correspondence: karol_tucki@sggw.pl (K.T.); o.orynycz@pb.edu.pl (O.O.)

**Abstract:** An analysis was carried out for different thermodynamic cycles of power plants with air turbines. Variants with regeneration and different cogeneration systems were considered. In the paper, we propose a new modification of a gas turbine cycle with the combustion chamber at the turbine outlet. A special air by-pass system of the combustor was applied and, in this way, the efficiency of the turbine cycle was increased by a few points. The proposed cycle equipped with a regenerator can provide higher efficiency than a classical gas turbine cycle with a regenerator. The best arrangements of combined air–steam cycles achieved very high values for overall cycle efficiency—that is, higher than 60%. An increase in efficiency to such degree would decrease fuel consumption, contribute to the mitigation of carbon dioxide emissions, and strengthen the sustainability of the region served by the power plant. This increase in efficiency might also contribute to the economic resilience of the area.

**Keywords:** thermodynamic cycle concepts; sustainability; modified cycle concepts; efficiency; energy systems

## 1. Introduction

Social and economic activity should aim to mitigate natural devastation. In recent years, the idea that improvements in quality of life should take place in greater harmony with the natural environment has gained popularity. Consequently, ecological problems have become the subject of reflection in various manufacturing sectors.

The development of production in various ecosystems should not only aim to keep up with demographic growth, but also to conserve equilibrium in the natural environment, so that it will be maintained in the best shape for the further development of future societies [1].

Mainly in reference to ecological politics, the definition of what is considered sustainable and durable development is essential. Ethical and economic values should be synchronized, aiming to match the premises of sustainable development. The term "sustainable development" is associated with many meanings. The multiplicity of this notion is reflected in various aspects and trends [2]. The  concept of sustainable development focusses attention on environmental, social, and economic factors, and clearly underlines the interdisciplinary character of the energy sector [3].

Sustainable energetics should take into account: (1) consumption and how to supply energy without exposing society to danger and (2) how to assure economic development whilst caring for the natural environment [4]. The energetic system is based on the notion of equalization, which should, among other things, consider the production of energy with regard to long-lasting economic and environmental goals [5].

Disseminating sustainable development on a global scale [6] is the focus of activities aiming to conceptualize how energetics could be developed through skillful conservation of energy on the local level. The sustainable management of these resources has to be supported by sustainable development of energy [7], and should deliver information that enables the evaluation of the energetics in ecological, economic, and social dimensions.

The formation of a green economy requires the union of transformation processes across the various regions of the country, along with the active participation of enterprises. One of the factors assuring the development of the modern economy is the uninterrupted delivery of energy. The energy resources trade, as well as industrial activity, is bound to the pollution of the atmosphere. This often leads to regional conflicts and ecological threats, even on a global scale. Knowledge of the suitability of fuels and the their suitable selection is indispensable to the safe and effective design as well as exploitation of technical objects [8].

Energetic systems require a deep understanding, taking into account the changes in production technologies [9]. Therefore energetics must occupy an important place amongst the different areas of sustainable development. The aim of the present paper is to highlight the possibility of utilizing circulation power stations in the context of sustainable development. Such a development should be implemented universally, with regard to the politics of individual states.

A problem of the national power generation sector is the relatively low efficiency of energy generation from coal, which is additionally accompanied by high emissions of carbon dioxide [10–12]. The average efficiency of Polish power plants is lower than elsewhere in the EU. The most efficient power plants in Poland are the newest units: Łagisza II (41% efficiency), Pątnów II (41% efficiency), Bełchatów II (42% efficiency), and Opole II (45% efficiency). Note that although these values can differ depending on the quoted source, the differences never exceed two percentage points. The majority of Polish power plants have an efficiency level below 36%, whilst for the oldest ones it can even be below 30%. For the present efficiency level, the assessed $CO_2$ emissions from steam power plants is approximately equal to 1100–1200 $kgCO_2$/MWh (for lignite-fired power plants). Modern power units operating at supercritical steam parameters ensure higher efficiency (48%–50% for hard coal-fired power plants), which, however, is decreased by more than ten percentage points by the unavoidable use of $CO_2$ sequestration systems. It is also noteworthy that several Polish power plants, with a total installed power capacity of about 6 GW, are likely to be closed during the next few years, due to poor technical condition [13–16].

*Problem Posing*

In the case of distributed power generation, the efficiency of small electric power plants is even lower. Organic Rankine Cycle (ORC)-based thermal power plants have an efficiency as low as approximately 12%. At present, one of the most advanced and efficient electric power plants is a power unit planned for construction in the USA, which forms part of the framework of a program financed by the US Department of Energy (US DOE). This unit will operate at ultra-supercritical steam parameters (with a pressure of about 35 MPa and temperature up to 760 °C), and its efficiency is expected to reach approximately 55% [17,18]. Such high steam pressures and temperatures require special materials and technologies, the use of which in Poland remains confined to a rather distant future [19]. At present, the highest efficiencies are reached in combined-cycle power plants (slightly over 60%, designed by Siemens), but these power plants are not coal-fired [20–22]. Hence, the question arises as to which direction attempts should be made to increase the efficiency of electric power generation in Poland. This can be reached by increasing the thermal efficiency of the applied cycle and/or by increasing

the efficiency of its individual components. In the case of the most recent and most technologically advanced large steam power plants, the efficiencies of their machinery and equipment are very high and we cannot expect that they can be further significantly increased [23–28]. For instance, the maximal efficiency of boilers ranges within 92–95%, high-pressure turbines 88–94%, medium-pressure turbines 90–97%, low-pressure turbines 88–95%, electric current generators 98.5–99%, and water pumps—about 85%. This is complemented by low losses in external glands and low mechanical losses in turbine sets, which are clearly lower than one percentage point. Therefore, the other option (i.e. increasing the thermodynamic cycle efficiency) seems more promising.

## 2. Materials and Methods

The topic of the present study concerned the possibility of increasing the energy efficiency of small turbines working at various structural configurations.

Thermodynamic analysis was applied to each technical configuration and this formed the main methodology applied in the current research. Five configuration variants were analyzed. Figure 1 shows five different configurations of gas turbine sets:

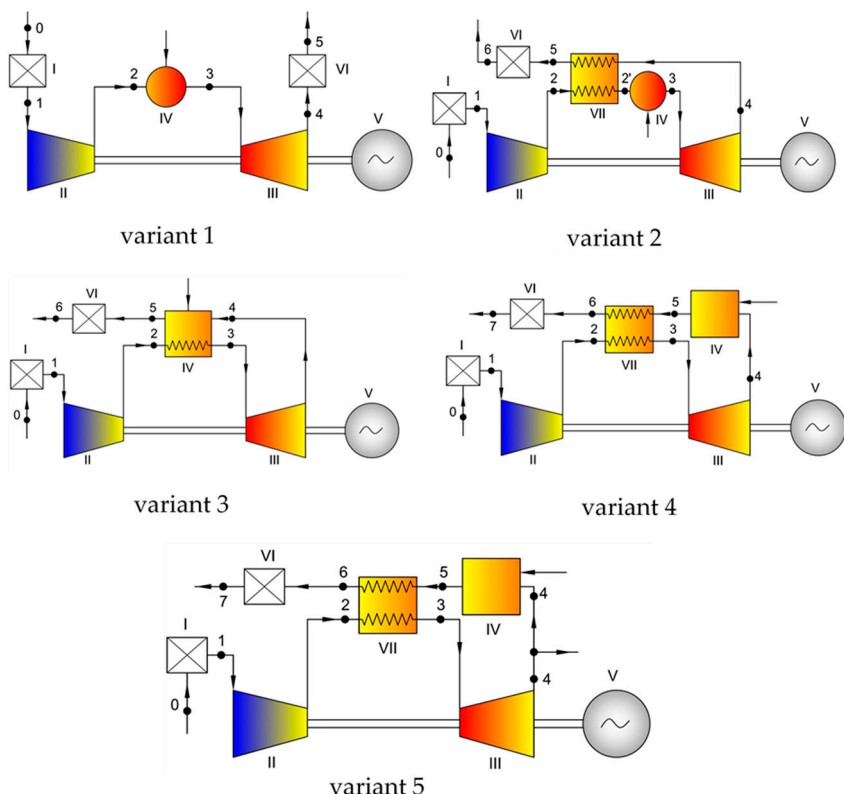

**Figure 1.** Analyzed turbine set arrangements. Variant 1: turbine set operating according to the simple open cycle; Variant 2: turbine set operating according to the open cycle with a regenerator; Variant 3: turbine set operating according to the open cycle with a combustion chamber at the turbine exit [29–31]; Variant 4: turbine set operating according to the open cycle with an external combustion chamber at the turbine exit and a high-temperature heat exchanger; Variant 5: turbine set operating according to the open cycle with partial bypassing of the external combustion chamber at the turbine exit and with a high-temperature heat exchanger.

In regenerator VII (variant 4), the exhaust gasses had a higher specific heat and a higher mass flow rate than the warmed up air. Thus, the temperature difference $T_5 - T_6$ was lower than $T_3 - T_2$. In variant 5, by drawing off some air, we decreased outlet temperature $T_6$ and reduced fuel consumption, as well as increasing cycle efficiency.

## 3. Results

Generally, the efficiency of the thermodynamic cycle is given by the well-known formula: $\eta = 1 - T_d/T_g$, where $T_d$ and $T_g$ represent the average temperatures of the hot reservoir (from which the heat is taken) and the cold reservoir (to which it is supplied), respectively. The highest possible temperature $T_g$ and the lowest possible temperature $T_d$ can be reached by the use of regeneration, as seen in Figure 2 (this figure shows the Brayton cycle [32], but the above is true for an arbitrary closed thermodynamic cycle).

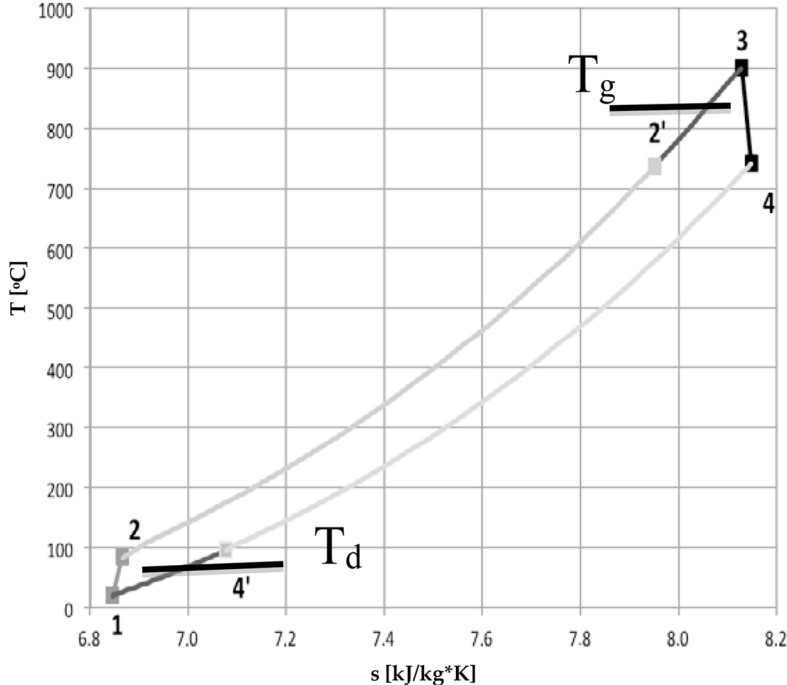

**Figure 2.** Cycle with regeneration (sample case).

The key role in regeneration is played by the so-called regeneration ratio $\varepsilon$ (sometimes also referred to as the regenerator efficiency), which is defined as the ratio of the actual executed temperature increase to its theoretical maximal value for the heated medium, $\varepsilon = (T_{2'} - T_2)/(T_4 - T_2)$ (Figure 2). It is well-known that for the cycle of a gas turbine set with a regenerator, increasing the regeneration ratio $\varepsilon$ leads to an increase in the cycle efficiency and to a decrease in the optimal compression value. Assuming that $\varepsilon = 1$, the efficiency of the ideal cycle of a gas turbine with a regenerator approaches the Carnot cycle efficiency when the compression approaches 1. However, in practice, the regeneration ratio has not exceeded $\varepsilon = 0.8$–$0.85$ for a long time, as increasing $\varepsilon$ leads to an extreme increase of the regenerator heat transfer surface, which increases proportionally to $\varepsilon/(1 - \varepsilon)$. The next limitation was the maximal permissible temperature for regenerator material. The heat exchangers offered by the producers have a maximal temperature limit of approximately 700 °C. In recent years, there has been changes in this area, as a result of studying the use of $CO_2$ for cooling high-temperature gas reactors and developing the technology for production of so-called ceramic heat exchangers. At present, the production technology of heat exchangers operating at $\varepsilon = 0.98$ and at medium temperatures equal to 900–1000 °C, or even exceeding 1200 °C in the case of ceramic heat exchangers, is considered fully mature [33–38]. Adopting solutions of this type provides new opportunities for increasing the efficiencies of gas turbine sets and combined systems. The possibility of using highly efficient high-temperature heat exchangers has renewed the interest in closed gas cycles and gas turbine sets with a combustion chamber at the turbine exit.

## 4. Discussion

Within the analyzed variants, the variant that revealed the lowest efficiency (for the same assumed turbine inlet temperature and the same individual efficiencies of turbine set components) was Variant 1 (i.e., simple open cycle). The efficiency of Variant 2 (i.e., cycle with regenerator) was higher by a few percentage points. Even higher efficiency was reached by Variant 3, in which the air flowed to the combustion chamber directly from the turbine exit, which corresponded to a cycle with a regenerator of 100% efficiency, $\varepsilon = 1$. Variant 4 was intended for systems in which, in practice, arbitrary fuel can be combusted. However, due to the limited temperature difference $T_5 - T_3$, its efficiency was close to that in Variant 2. In Variant 5, part of the air leaving the turbine bypassed the combustion chamber, which improved the efficiency and provided wider opportunities for the use of a combined system. However, it should be noted that the heat exchange area of the regenerators in Variants 2, 4, and 5 were approximately 55–70% higher than in Variant 3.

The efficiency analysis was performed for large power turbine sets, at the assumed relatively high efficiencies of individual elements. In particular, the assumed efficiencies of the turbine and compressor were equal to 90%, while they were 98% for both the electric current generator and the combustion chamber. The assumed design values and working media parameters for particular points of the optimized cycles are shown in Tables 1 and 2, respectively.

**Table 1.** Assumed design parameters.

| Parameter | Unit | Value | |
|---|---|---|---|
| $\eta_{compressor}$ | [-] | 0.900 | |
| $\eta_{turbine}$ | [-] | 0.900 | |
| $\eta_{mech}$ | [-] | 0.980 | |
| $\eta_{leakage}$ | [-] | 0.980 | |
| $\eta_{Generator}$ | [-] | 0.980 | |
| $\eta_{comb.cham}$ | [-] | 0.980 | |
| $p_i/p_{i-1}$ | [-] | 0.995 | air inlet duct/filter |
| $p_i/p_{i-1}$ | [-] | 0.995 | exhaust gases duct/filter/silencer |
| $p_i/p_{i-1}$ | [-] | 0.99 | combustion chamber |
| $p_i/p_{i-1}$ | [-] | 0.99 | regenerator |
| LHV | [MJ/kg] | 24 | |

The relative efficiency values obtained for the analyzed variants after assuming the turbine inlet temperature $T_3 = 900\ ^\circ$C are shown in Figure 3 with Variant 1 as the reference. They confirm the above observations. Figure 4 shows optimal compression values for the analyzed variants. The optimum values of compression ratio for each case were obtained as a result of the calculation of a large number of different variants. Note that at high efficiency ($\varepsilon$) of heat exchangers, the compression values in the analyzed systems were extremely low. On the one hand, this would make the turbine and compressor structures simpler and cheaper to produce, but on the other hand would decrease the specific power of the turbine set.

The efficiency values were heavily affected by the turbine inlet temperature and the efficiencies of heat exchangers. For instance, decreasing the final temperature difference $T_5 - T_2$ from 50 $^\circ$C to 10 $^\circ$C in Variant 3 increased the efficiency by about 9.5%. The highest efficiencies were obtained for combined gas–steam systems (see Figure 5). In those cases, exceeding 60% efficiency became possible even when the air turbine inlet temperature was as low as $T_3 = 900\ ^\circ$C. Figure 6 shows a sample case of a compressor and an air turbine cooperating in the combined power plant, with an output power of about

55 MW. Meanwhile, Figure 7 presents the flow part of the steam turbine for this power plant. Even higher efficiencies could be reached in combined cycles consisting of a closed gas system and a steam cycle.

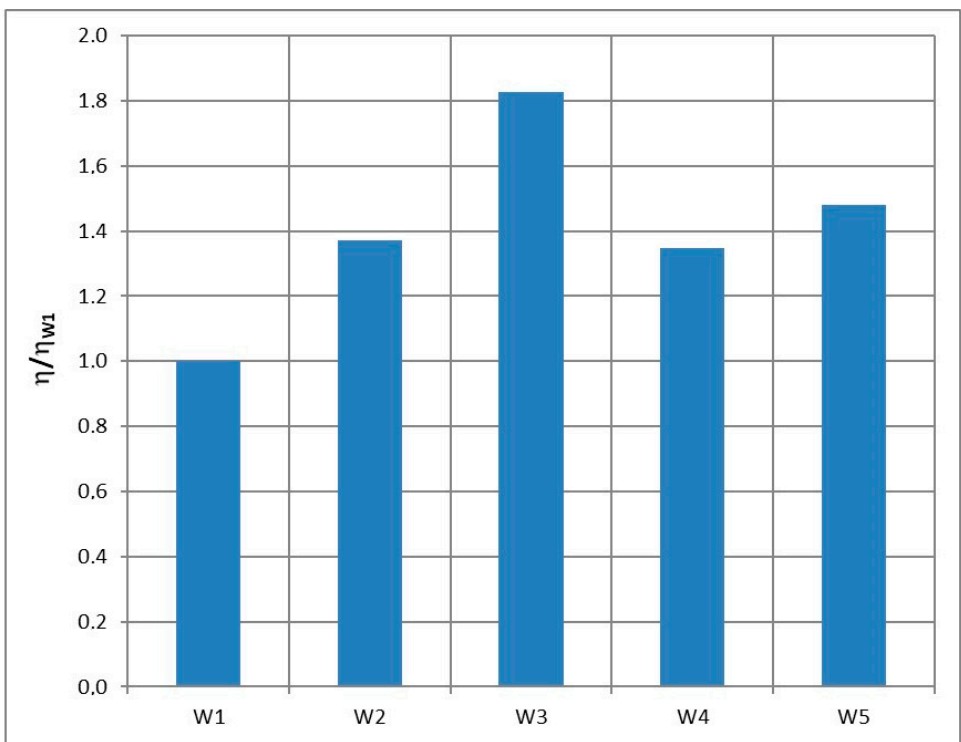

**Figure 3.** Relative frequencies of analyzed turbine sets.

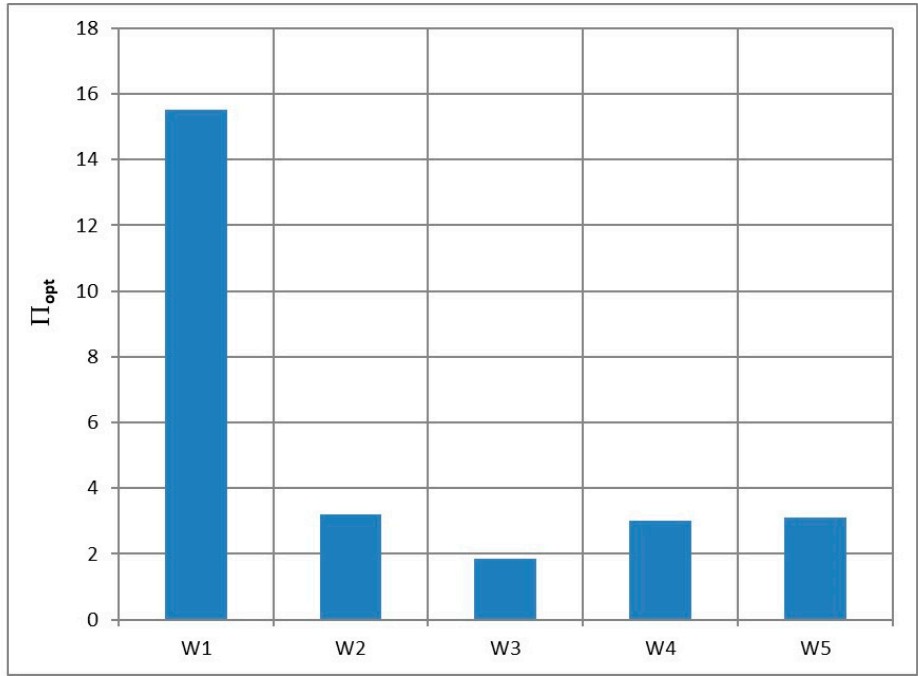

**Figure 4.** Optimal compression values of analyzed turbine sets.

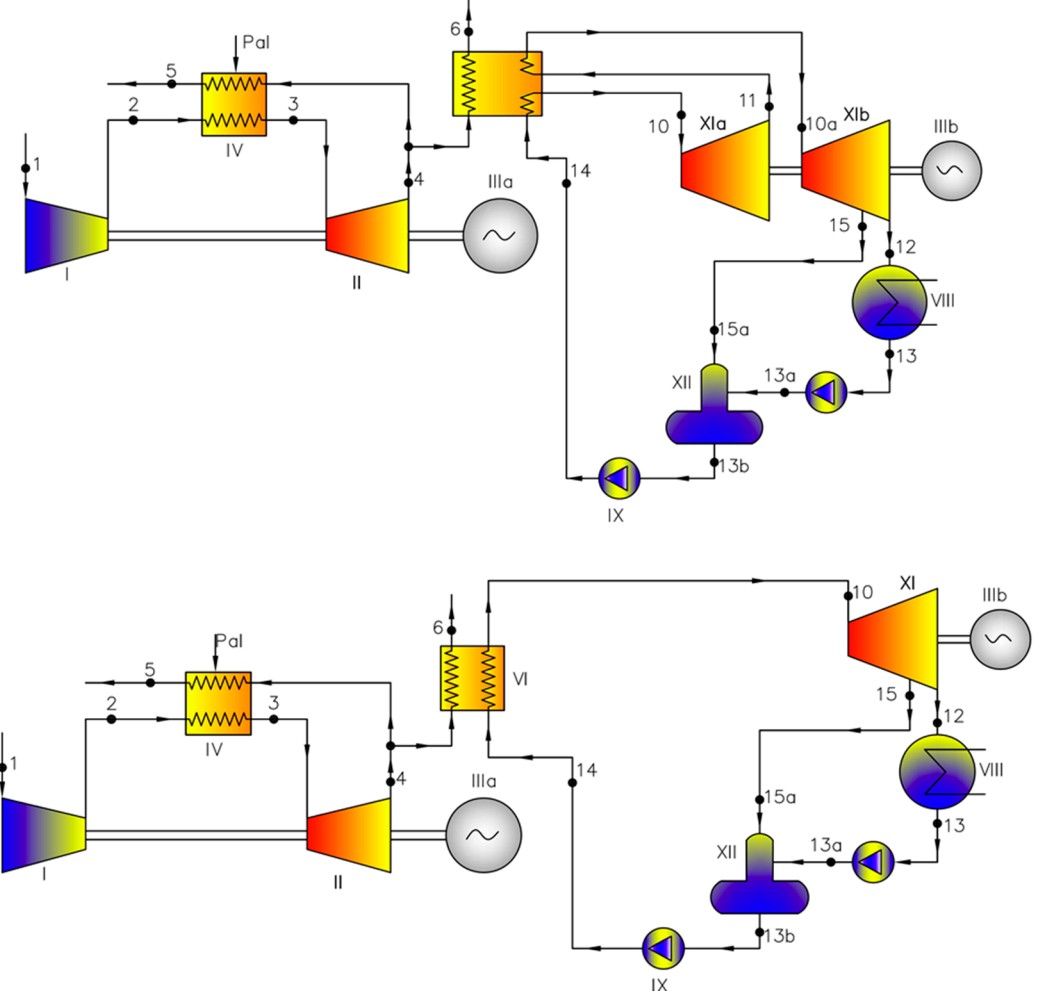

**Figure 5.** Combined steam–air system (sample solutions).

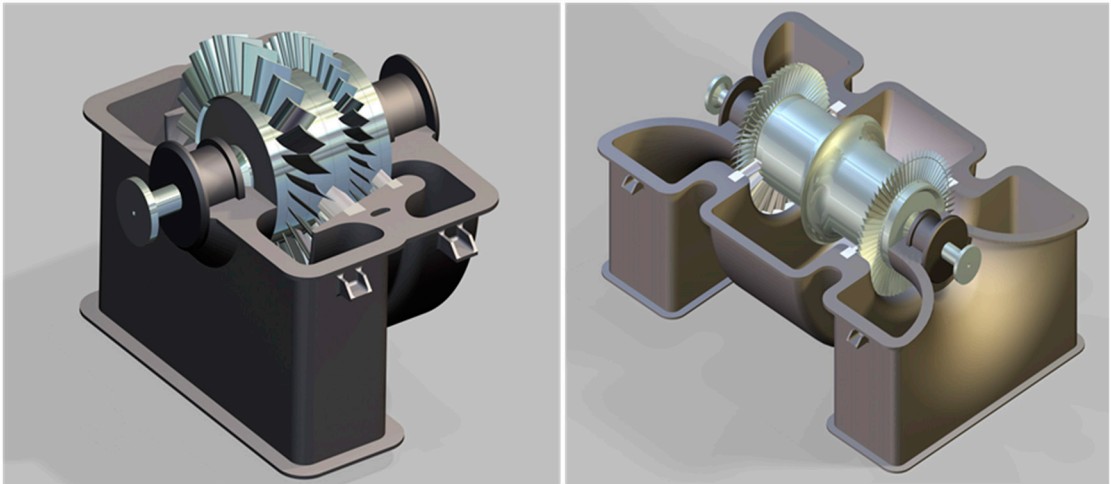

**Figure 6.** Compressor (**left**) and air turbine (**right**) for 55 MW combined steam–air system.

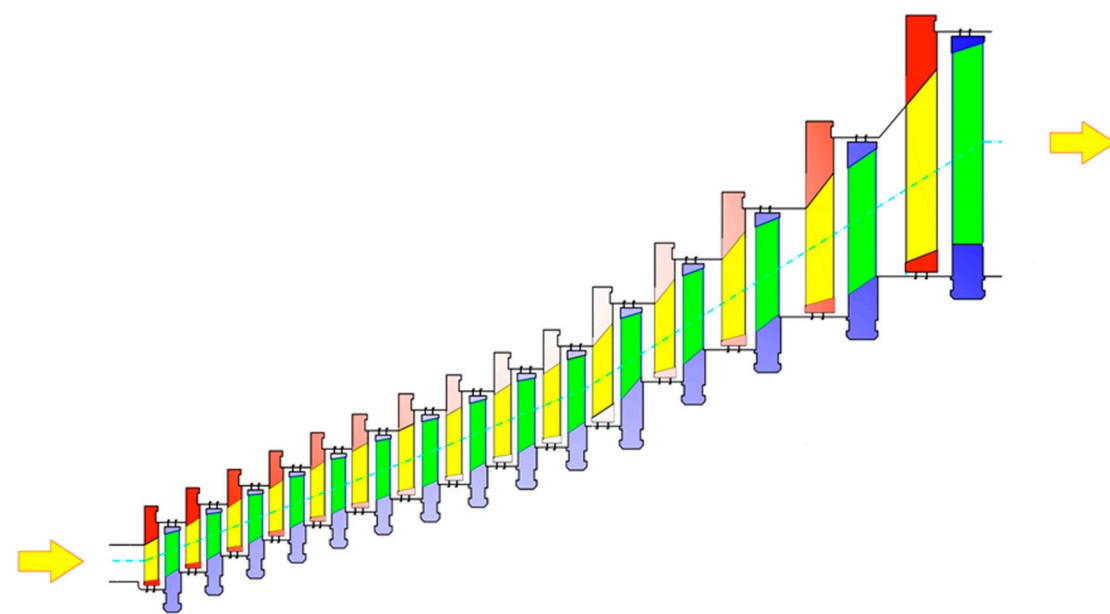

**Figure 7.** Schematic of steam turbine flow part for 55 MW combined steam–air system.

**Table 2.** Working media parameters in particular cycle points.

|  |  | **W1** | **W2** | **W3** | **W4** | **W5** |
|---|---|---|---|---|---|---|
| **Π** | [-] | 15.50 | 3.20 | 1.85 | 3.00 | 3.10 |
| **$p_0$** | [MPa] | 0.1000 | 0.1000 | 0.1000 | 0.1000 | 0.1000 |
| **$p_1$** | [MPa] | 0.0995 | 0.0995 | 0.0995 | 0.0995 | 0.0995 |
| **$p_2$** | [MPa] | 1.5423 | 0.2985 | 0.1841 | 0.2985 | 0.3085 |
| **$p_{2'}$** | [MPa] | - | 0.2955 | - | - | - |
| **$p_3$** | [MPa] | 1.5268 | 0.2926 | 0.1822 | 0.2955 | 0.3054 |
| **$p_4$** | [MPa] | 0.1005 | 0.1015 | 0.1015 | 0.1025 | 0.1025 |
| **$p_5$** | [MPa] | 0.1000 | 0.1005 | 0.1005 | 0.1015 | 0.1015 |
| **$p_6$** | [MPa] | - | 0.1000 | 0.1000 | 0.1005 | 0.1005 |
| **$p_7$** | [MPa] | - | - | - | 0.1000 | 0.1000 |
| **$T_0$** | [°C] | 20 | 20 | 20 | 20 | 20 |
| **$T_1$** | [°C] | 20 | 20 | 20 | 20 | 20 |
| **$T_2$** | [°C] | 407.04 | 148.40 | 82.59 | 140.11 | 144.30 |
| **$T_{2'}$** | [°C] | - | 633.24 | - | - | - |
| **$T_3$** | [°C] | 900 | 900 | 900 | 900 | 900 |
| **$T_4$** | [°C] | 381.71 | 643.24 | 737.47 | 624.46 | 617.19 |
| **$T_5$** | [°C] | 381.71 | 199.89 | 92.59 | 910 | 910 |
| **$T_6$** | [°C] | - | 199.89 | 92.59 | 221.89 | 154.30 |
| **$T_7$** | [°C] | - | - | - | 221.89 | 154.30 |

In those solutions, the expected efficiencies could exceed 60% (60%–65%) or even 65% at higher air turbine inlet temperatures $T_3$ = 1000–1200 °C. Note that these temperatures are lower than those currently recorded in the most technologically advanced gas turbine sets.

### 5. Conclusions

Logging and husbanding energy constitute the essential elements of sustainable development. Laying the foundations for sustainable development relates to the energetics upon which many social and economic processes depend. The achievement of sustainable energetics should be the object of further investigations. As a future direction, Poland should aim to enlarge its ecological consciousness with the implementation of sustainable energetics. The proposed creation of circulation-combined power stations is one of the possible solutions to the problems of the Polish coal industry and the national energy production sector. It might also bring closer wide-ranging changes in the energetic politics.

The proposed combined power plant cycle is a solution variant for the problems faced by the Polish coal-based energy production sector. It would create opportunities for building highly efficient coal power plants (>60% efficiency). It would also respond to Poland's need to meet environmental protection requirements, as high efficiency will make $CO_2$ capturing and storage unnecessary, while simultaneously ensuring the high profitability of energy generation from coal.

The proposed technology is distinguished by:

- The original scheme of the thermodynamic cycle of power plant;
- The low parameters of the working medium;
- Its high efficiency, equal to about 60%–65% (or more). This means it would be possible to nearly double the amount of electric energy generated from national coal (for a given amount of coal, as compared to the present state of technology). In practice, this would be equivalent to decreasing fuel consumption by half. It would also be accompanied by a smaller consumption of cooling water (by at least 2–4 times) and smaller amounts of heat being released to the environment;
- The fuel used can be black or brown coal, resources which are plentiful in Poland. This solution will also allow for diversification, that is, the possible use of other fuels such as oil, natural gas, biofuels, biomass, bio gas, wooden pellets, and/or agricultural and municipal waste;
- Lower power plant construction costs, compared to, for instance, both subcritical and supercritical steam power plants, or modern combined gas-steam systems;
- Cheaper power plant operation and maintenance. Flow parts of the compressor and turbine would remain clean during the entire useful life of the power plant, as they would not be polluted with exhaust and would not require repairs and cleaning as in other highly efficient solutions;
- Possibility of application in thermal and electric power plants, in cogeneration and trigeneration systems (i.e., for the simultaneous production of electricity, heat, and useful cold);
- The absence of carbon dioxide capture and storage installation, which significantly decreases the power plant construction costs and avoids the considerable efficiency drop related to the operation of this installation;
- Halving of $CO_2$ emissions, down to below 600 $kgCO_2$/MWh (i.e., to a value which is comparable with the $CO_2$ emissions from present natural gas-fired power plants). Additionally, the emission of greenhouse gases would be reduced to half that of the emissions of present coal-fired devices. This solution would meet, with a surplus, all EU requirements (program 3×20), and would simultaneously be a rescue for Polish coal, which could then be considered as "clean fuel".

The proposed solutions would provide sales for Polish coal and would positively affect the energy security of the country. An optional solution to that described in the article is a combined system consisting of a closed $CO_2$ cycle and steam cycle. Design calculations performed for this variant and preliminary designs of gas and steam turbines for a 335 MW power plant confirmed the possibility of its implementation and of reaching the "60% plus" efficiency.

**Author Contributions:** Conceptualization, K.K. and M.P.; R.S.; W.W.; Methodology, K.T. and O.O.; Validation, K.K. and M.P.; A.B.; Investigation, R.S. and W.W.; Writing—Original Draft Preparation, K.T. and O.O.; A.B.; Funding Acquisition, K.K.

**Funding:** This research received no external funding.

**Acknowledgments:** The Authors wish to express their deep gratitude to Gdansk University of Technology for financial support given to the present publication (Krzysztof Kosowski) The research contributing to the present publication, conducted by Olga Orynycz, has been performed under the financial support No. S/WZ/1/2015 financed by the Polish Ministry of Science and Higher Education from the funds dedicated to science.

**Conflicts of Interest:** The authors declare no conflict of interest. The funders had no role in the design of the study; in the collection, analyses, or interpretation of data; in the writing of the manuscript, and in the decision to publish the results.

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
