# Peer review of "Thermodynamic Cycle Concepts for High-Efficiency Power Plans. Part A: Public Power Plants 60+"

_sustainability, doi:10.3390/su11020554_

Round 1

Reviewer 1 Report

In reviewing the manuscript, several comments come to mind.

1.       The paper is very innovative and does address the role that thermodynamic efficiency plays in reducing CO2 emissions.  The authors have truly shown how clean coal can be compatible with reducing carbon emissions without excessive sequestration. I would highly recommend its publication.

2.       The first point that would be useful to clarify is the difference between Variant 3 and Variant 4 as to how the combustion of the fuel is handled.  Variant 3 appears to have the combustion going on in the recuperative heat exchanger.  This would appear to be a new technology and its developmental level should be referenced or at least described in more detail.

3.       Any regenerative cycle requires a large heat transfer surface dedicated to the regenerator.  Some consideration of relative sizes for the 5 variants would be useful. 

4.       The statements made concerning ceramic heat exchangers in lines 134 to 142 should be referenced as the maturity of such systems is not obvious.

5.       Some comment should be made about why drawing off part of the working fluid in Variant 5 produces a higher efficiency than using the full working fluid in Variant 4.  That is unless the drawn off working fluid is used to drive a bottoming cycle.  If this is the case, it should be stated.

6.       Even with the low compression ratios of Variants 2 through 5, a split compressor with an intercooler should improve the overall cycle efficiencies.  This technology is more common than the combined combustion chamber-heat exchanger of Variant 3.

7.       It is not clear that Figure 7 adds anything to the manuscript.

8.       It would be useful if all components referencing power levels indicated whether they are thermal power levels or electrical power levels of a supposed 99% efficient generator.

Author Response

Dear Reviewer,

Thank you very much for your review containing important remarks.
We tried whenever it was possible, to consider all your suggestions.

With the respect

Reviewer 2 Report

REVIEW

Manuscript number: sustainability-424187

Title: Thermodynamic cycle concepts for high-efficiency power plans. Part A: Public power plants 60+

Authors: Krzysztof Kosowski, Karol Tucki, Marian Piwowarski, Robert Stępień, Olga Orynycz, Wojciech Włodarski, Anna Bączyk

The paper contains the analysis of different thermodynamic cycles of Power plants with air turbines. The subject and content of this paper complies with the purpose of Sustainability - the International Journal.

The abstract has state briefly the purpose of the research, the principal results and major conclusions. It allows the reader to easily read the content of the article. The article has 5 keywords.

The article has no nomenclature. The Authors use a huge number of abbreviations, acronyms and markings. I suggest adding the nomenclature at the beginning of the paper, which will help the reader to decipher individual abbreviations.

The structure of the paper contains 7 figures, 0 tables and 0 equations.

The literature in paper has 29 references and is appropriately selected. Each has its own reference in the manuscript.

The way of explaining the methodology are not good. There are no assumptions characteristic for gas turbine operation – air and fuel (more information about it in essential remarks). Drawings made in poor graphic quality.

The paper also has interesting elements. In particular, the results and discussion section looks interesting. Nevertheless, the paper also has some drawbacks and points to be revised as follows.

EDITORIAL REMARKS

·         Introduction; Line 74 and 75: “1100-1200 kgCO2/MW h” should be “1100-1200 kgCO2/MWh”. The same situation at line 245.

·         “Problem posing” section; Line 87: “up to 7600C” should be “up to 760°C”. The same situation at line 134, 138, 164, 180, 183 and 188.

·         “Problem posing” section; Line 87: “35MPa” should be “35 MPa”.

·         “Material and Methods” section; Figure 1; No description of points I, II, III, IV, V, VI and VII.

·         “Conclusions” section; Line 222; “CO2capturing” should be “CO2 capturing”.

ESSENTIAL REMARKS

·         In Introduction section Authors wrote: “Modern power units operating at supercritical steam parameters ensure higher efficiency (48%-50%)…”.

I agree but only for hard coal fired power plant. For lignite fired power plant efficiency is max. 42%.

·         In Introduction section Authors wrote: “For present efficiency level, the assessed CO2 emissions from steam power plants are approximately equal 1100 – 1200 kgCO2/MW h”. I agree but only for lignite fired power plant. For hard coal fired power plant emission of CO2 is around 870 kg/MWh (Net electric efficiency ηel.n = 0.48; LHV = 24 MJ/kg; The content of carbon in fuel c = 0,76; MCO2 = 44 kg/kmol; MC = 12 kg/kmol; eCO2 = (c * (MCO2/MC) * 3600)/(LHV * ηel.n) = 870,83 kg/MWh).

·         In “Problem posing” section Authors wrote: “Nowadays, the highest efficiencies are reached in combined gas-steam systems (slightly over 60%, Siemens), but these power plants are not coal-fired [20-22].

First of all I suggest to use “combined cycle power plants” instead of “combined gas-steam systems”. I would just like to point out that the reported efficiency is not the same efficiency that is presented by the Authors in section 4. The efficiency cited is the electrical efficiency of electricity production - not efficiency of thermodynamic cycle.

·         In Figure 4, the Authors showed the optimum gas turbine compression ratio for individual variants. No information on how the optimization looked. How did the Authors obtain these optimal compression ratios?

·         There are no assumptions characteristic for gas turbine operation. No information about fuel (fuel composition, parameters, LHV). No information about air (air composition, parameters). No information about isentropic and mechanical efficiencies of compressor and expander. What are the pressure losses on individual elements? If there is a different compression ratio in each variant, what does the change in the isentropic efficiency of individual elements look like?

·         The Authors, in the introduction, cite net electrical efficiency of power plants, in the analysis should also show the net electrical efficiency of the tested variants. The authors should also show the air / fumes parameters at characteristic points of the analyzed variants. It is also interesting to show the outlet loss in the analyzed variants.

·         Placing the combustion chamber at the outlet of a gas turbine causes an increase in the outlet loss. In the case of a combined system, this would mean that it would be necessary to use steam supercritical parameters in the steam cycle. This means that while in my humble opinion the concept is interesting, it will never be commercialized, because there are no steam cycles with heat recovery steam generator for supercritical parameters. Why? Because this solution is 4 times more expensive than the modernization of the gas turbine itself, which was also reported by Polish scientists in the Kotowicz J., Job M., Brzęczek M., The characteristics of ultramodern combined cycle power plants. Energy 2015;92:197-211.

In view of the above, The article has high scientific values and a modern approach to the subject after making corrections.

Author Response

Honorable Reviewer

Thank you very much for your review containing important remarks.
We tried whenever it was possible, to consider all your suggestions.

With the best regards, and thanks

Round 2

Reviewer 2 Report

Accept in present form.